

# Response of *Escherichia coli* minimal *ter* operon to UVC and auto-aggregation: pilot study

Lenka Jánošíková[1], Lenka Pálková[2], Dušan Šalát[1], Andrej Klepanec[1] and Katarina Soltys[3,4]

[1] Faculty of Health Sciences, University of St. Cyril and Methodius in Trnava, Trnava, Slovak Republic
[2] Medirex group, Bratislava, Slovak Republic
[3] Department of Microbiology and Virology, Faculty of Natural Sciences, Comenius University in Bratislava, Bratislava, Slovak Republic
[4] Comenius University Science Park, Comenius University in Bratislava, Bratislava, Slovak Republic

## ABSTRACT

**Aim**. The study of minimal *ter* operon as a determinant of tellurium resistance ($Te^R$) is important for the purpose of confirming the relationship of these genes to the pathogenicity of microorganisms. The *ter* operon is widespread among bacterial species and pathogens, implicated also in phage inhibition, oxidative stress and colicin resistance. So far, there is no experimental evidence for the role of the *Escherichia coli (E. coli)* minimal *ter* operon in ultraviolet C (UVC) resistance, biofilm formation and auto-aggregation. To identify connection with UVC resistance of the minimal *ter* operon, matched pairs of Ter-positive and -negative *E. coli* cells were stressed and differences in survival and whole genome sequence analysis were performed. This study was aimed also to identify differences in phenotype of cells induced by environmental stress.

**Methods**. In the current study, a minimal *ter* operon($terBCDE\Delta F$) originating from the uropathogenic strain *E. coli* KL53 was used. Clonogenic assay was the method of choice to determine cell reproductive death after treatment with UVC irradiation at certain time intervals. Bacterial suspensions were irradiated with 254 nm UVC-light (germicidal lamp in biological safety cabinet) in vitro. UVC irradiance output was 2.5 mW/cm$^2$ (calculated at the UVC device aperture) and plate-lamp distance of 60 cm. DNA damage analysis was performed using shotgun sequencing on Illumina MiSeq platform. Biofilm formation was measured by a crystal violet retention assay. Auto-aggregation assay was performed according to the Ghane, Babaeekhou & Ketabi (2020).

**Results**. A large fraction of Ter-positive *E. coli* cells survived treatment with 120-s UVC light (300 mJ/cm$^2$) compared to matched Ter-negative cells; $\sim$5-fold higher resistance of Ter-positive cells to UVC dose ($p = 0.0007$). Moreover, UVC surviving Ter-positive cells showed smaller mutation rate as Ter-negative cells. The study demonstrated that a 1200-s exposure to UVC (3,000 mJ/cm$^2$) was sufficient for 100% inhibition of growth for all the Ter-positive and -negative *E. coli* cells. The Ter-positive strain exhibited of 26% higher auto-aggregation activities and was able to inhibit biofilm formation over than Ter- negative strain (**** $P < 0.0001$).

**Conclusion**. Our study shows that Ter-positive cells display lower sensitivity to UVC radiation, corresponding to a presence in minimal *ter* operon. In addition, our study suggests that also auto-aggregation ability is related to minimal *ter* operon. The

Corresponding author
Lenka Jánošíková,
lenka.janosikova@ucm.sk

role of the minimal *ter* operon (*terBCDE∆F*) in resistance behavior of *E. coli* under environmental stress is evident.

## INTRODUCTION

Little is known about the biochemical activities and biological function of the TerB, TerC, TerD and TerE proteins that are determined by genes of the minimal *ter* operon encoded by a plasmid. To this date, there is no study of resistance to UVC radiation, biofilm formation and auto-aggregation in tellurium resistance (Te[R]) *E. coli* cells.

Ultraviolet (UV) irradiation is electromagnetic irradiation, where the UVC spectrum, is absorbed by the nucleic acids (*Gurzadyan, Görner & Schulte-Frohlinde, 1995*) producing several types of damage that interfere among other with replication and transcription of DNA. UV light promot a major and minor photoproduct formation. The major photoproducts of DNA are cyclobutane pyrimidine dimers (CPDs) and pyrimidine-(6,4)-pyrimidone photoproducts (6-4PPs) (*Tropp, 2008*), where covalent bonds are present between adjacent pyrimidines on the same DNA strand (*Pirnie, Linden & Malley, 2006*). Moreover, the minor photoproducts of DNA are Dewar photoproducts, TA* photoproducts, AA* photoproducts and Porschke photoproducts with a very small quantum yield production (*Siede, Kow & Doetsch, 2005*). Other types of UV damage are also known, such as protein-DNA cross-links, DNA-DNA cross-links, single strand breaks and double strand breaks. However, pyrimidine dimers are the most common damage resulting from UV light (*Pirnie, Linden & Malley, 2006*). If UV-induced damage is not repaired or eliminated from DNA, it may lead to mutagenesis, cellular transformation, and cell death. Even if the lesion is removed, the result can be a mutation (*Tropp, 2008*). This was the reason why we started looking for some connection between the potential increased UVC resistance and mutation rate of the cells.

It is well known that UV radiation is used to inactivate microorganisms, i.e., UV disinfection. Ultraviolet disinfection can be used to disinfect air, water, wastewater, laboratory equipment, medical instruments, food and beverages. Therefore, it is extremely important to study the phenomenon of UVC resistance. *Dai et al. (2012)* discussed the potential of UVC irradiation as a different concept to standard methods used to treat localized infections. The gradual emergence of populations of antibiotic-resistant bacteria has become a major public health problem. Antibiotic resistance has led to the search for alternative antimicrobial approaches to which, microorganisms will not be easily able to develop resistance. On the other hand, the existence of UVC-resistant microorganisms and the resistance mechanisms themselves should be considered. For instance, the multiple extremes resistant bacterium *Deinococcus radiodurans* (*D. radiodurans*) is able to withstand harsh condition, such as UV radiation (*Ott et al., 2017*).
*Burian et al. (1990)* identified during testing of the group of clinical isolates a uropathogenic strain of *E. coli* KL53 resistant to tellurite. The *ter* operon (terXYW and terZABCDEF) has been found on a large conjugative plasmid pTE53 and the 5 kb region of *ter* operon was cloned into the low copy plasmid pACYC184. This plasmid contained minimal *ter* operon and was marked as pLK18 (*Burian et al., 1998*), which we used in our work. The genes from the large conjugative plasmid pTE53 of *E. coli* strain KL53: *terB*, *terC*, *terD* and *terE* are essential for conservation of the Te$^R$ whereas the gene *terF* is not important in this respect (*Kormutakova, Klucar & Turna, 2000*). It is important to note that homologous genes have been found in bacteria like *E. coli* O157:H7, *D. radiodurans* (*Taylor et al., 2002*), *Shigella flexneri*, *Yersinia pestis* (*Taylor, 1999*), *Klebsiella pneumoniae* (*Chen et al., 2004*), *Vibrio cholerae*, *Proteus mirabilis* (*Toptchieva et al., 2003*), etc. of which the vast majority are pathogenic microorganisms and cause serious diseases in humans worldwide (*Lim, Yoon & Hovde, 2010*; *Schroeder & Hilbi, 2008*; *Demeure et al., 2019*; *Struve & Krogfelt, 2004*; *Nelson et al., 2009*; *Chen et al., 2012*).

It is assumed that the TerC–TerB complex can be associated with biochemical activities of proteosynthesis, C4-dicarboxylate transport, ATPase/chaperone activity and inner membrane stress response in bacteria (*Turkovicova et al., 2016*). On the other hand, TerC from *Arabidopsis thaliana* (Eukaryote) plays a crucial role in prothylakoid membrane biogenesis and thylakoid formation in early chloroplast development (*Kwon & Cho, 2008*). According to *Grant & Tsang (1990)*, TerD and TerE proteins share massive similarity with a cyclic AMP binding protein from *Dictyostelium discoideum* (Eukaryote). The results suggest that TerD family of eukaryotic homologs of TerD protein can bind soluble ligands, such as cAMP. Contrarily, a structural study from *Klebsiella pneumoniae* showed that TerD binds Ca (2+). These results suggest that some form of Ca (2+) signaling plays a crucial role in Te$^R$ (*Pan et al., 2011*). Evidence from the *Streptomyces coelicolor* suggests that this domain plays an important role in the adaptation of redox stress and calcium homeostasis in bacteria (*Daigle et al., 2015*). However, can all these known and unknown properties of Ter proteins (TerB, TerC, TerD and TerE) have an effect on the increased UVC resistance of cells?

The characteristic phenotype of cells bearing *ter* genes (*terZ, terA, terB* and *terC*) reported as influencing the potassium tellurite resistance present on the IncHI2 plasmid R478 in *E. coli,* is a filamentous morphology, indicated inhibition of cell division (*Whelan, Sherburne & Taylor, 1997*). Similarly, *Ponnusamy & Clinkenbeard (2015)* observed that *terZAB* genes mediated filamentous cellular morphology of *Yersinia pestis,* during macrophage infections, whereas *terCDE* confers tellurite resistance. These studies led to speculate that these genes are part of a bacterial adaptive strategy which also includes morphological changes of the cell to associated stress. In our previous study, we found up-regulation of the tryptophanase in Ter-negative (control) cells at a "sub-lethal" concentration of tellurite 3.9 μmol.l$^{-1}$ (*Aradska et al., 2013*). According to a study by *Kuczyńska-Wiśnik et al. (2010)* due to oxidative stress and over-expression of tryptophanase, which catalyzed indole synthesis, biofilm formation in *E. coli* was inhibited.

Therefore, we were interested in the connection between the Ter-positive *E. coli* cells and biofilm formation and/or auto-aggregation as a possible defense mechanism against stress.

The aim of this pilot study was also to analyze the survival of Ter-positive and -negative *E. coli* cells and determine the DNA damage (mutation rate) after three different doses of UVC radiation.

## MATERIALS & METHODS

### Bacterial strains and cultivation conditions

All bacterial strains and plasmids used in this study came from the collection of microorganisms of the Department of Molecular Biology, Faculty of Natural Sciences, Comenius University in Bratislava. The strains *E. coli* BL21 carrying plasmids pLK18 and pACYC184 were used for plasmid DNA preparation. Ter-positive strain *E. coli* BL21(DE3) bearing plasmid pLK18 and Ter-negative strain *E. coli* BL21 (DE3) carrying plasmid pACYC184 were used. Ter-positive as well as -negative cells were grown in LB medium supplemented with chloramphenicol. As an internal control of the biofilm formation methodology, we used the bacterium *Cronobacter malonaticus* 161007/29.

### Plasmid DNA preparation

Plasmid DNA was extracted from cells with "QIAprep Spin Miniprep Kit" (Qiagen, Hilden, Germany) according to manufacturer's instructions—Isolation from cells. The resulting DNA samples were visualized on a 1% agarose gel stained with Ethidium Bromide. Totally 100 ng of bacterial DNA quantified spectrophotometrically with NanoDrop 1000 Spectrophotometer (Thermo Fisher Scientific, Waltham, MA, USA) was used for electroporation.

### Preparation of competent cells

Overnight culture of *E. coli* BL21(DE3) was diluted 1:100 for growth in LB (Lauria-Bertani) broth. Cells were grown to an $OD_{600}$ 0.5 at 37 °C and were pelleted by centrifugation at 6,000 RPM for 10 min. Cells were resuspended in cold sterile $ddH_2O$. This treatment was repeated four times. After brief centrifugation, cell pellets were resuspended in 10% glycerol (0.3 mL). Finally, cells were transferred to microcentrifuge tube. For long-term storage of competent cells, cells were frozen immediately in a dry ice/ethanol bath and stored at −70 °C until needed.

### Electroporation

Electroporation was carried out using a BioRad gene Pulser. One hundred ng of DNA was mixed with electrocompetent cell and transferred to the cuvette. Electroporation conditions were as follows 125 Ω, 25 μF, 2.5 kV and 4,9 msec. After electroporation, 1mL LB was added to the cuvette. Cells were then transferred to culture tube for 1 h growth with shaking at 37 °C. Then, 0.1 mL of cells was spread on LB agar selective plates with chloramphenicol and incubated at 37 °C overnight.

## Ultraviolet C irradiation studies

Two-hundred microliters of an overnight culture Ter-positive and -negative cells were used to inoculate 20 mL of LB medium. When the $OD_{600}$ reached 0.6, cells grew for another 24 h to reach stationary growth phase. In the stationary growth phase, 5 mL solution was placed in sterile 55 mm diameter Petri dishes which were set on a shaking platform at 80 RPM. The cells were irradiated with 254 nm UVC-light (germicidal lamp in biological safety cabinet MSC-advantage $^{TM}$ Class II BSC Thermo Scientific) in an open (no lid on) media plates for 0 s, 120 s, 1200 s. UVC irradiance output was 2.5 mW/cm$^2$ (calculated at the UVC device aperture) and plate-lamp distance of 60 cm. Total exposure was calculated using the irradiance output multiplied by the amount of time they were irradiated. Immediately after irradiation, at each UVC dose (0.0 mJ/cm$^2$, 300 mJ/cm$^2$ and 3,000 mJ/cm$^2$), sample was serially diluted in LB medium with decimal dilution and 0.01 mL of 1:10,000 dilution was plated on LB agar with chloramphenicol. Cells were incubated on 37 °C for 24 h in dark and counted. For statistical reasons, three plates were needed for each dilution series. The following equation was used to calculate the number of Colony forming units (CFU) per mL from the original sample: CFU/mL = number of colonies/volume of culture plated (mL) × dilution factor. The triplicates were averaged and CFU were computed. Experiments were performed in biological "four replicates". Percentual survival was calculated using the following equation: Percentual survival = (CFU of exposed sample/ CFU of unexposed sample) × 100. The rest of the irradiated cells was collected by centrifugation (8,000 RPM), and stored as frozen pellets at −20 °C, to be used for DNA isolation and whole genome sequencing.

## Statistical analyses

All data for statistical analysis were obtained from minimum of three independent experiments. Averages and standard deviations were reported. Differences between experimental groups were set using unpaired t-Test: two-sample assuming equal variances or one-way ANOVA. $P$-values of 0.05 or less were considered statistically significant (*** $P < 0.001$, **** $P < 0.0001$). All statistical analyses were performed with GraphPad Prism 6.0 software (GraphPad Software Inc., San Diego CA, USA).

## Crystal violet assay for biofilm formation

Quantification of biofilm formation was assayed according to the published method with minor modifications (*Ghane, Babaeekhou & Ketabi, 2020*). Bacterial cells (Te-R and control) with $OD_{600} = 2$ were diluted with fresh LB medium (1:100) and 200 µl of diluted suspension was poured into the 96-well plates. The plates were incubated stationary at 37 °C for 24 h to create biofilm. Each well was gently rinsed 3 times with 150 µl sterile PBS, pH 7.2 to remove all planktonic bacteria. The attached cells were fixed with 150 µl methanol and dried up at room temperature. After that, 150 µl of 0.1% Crystal Violet solution (Sigma-Aldrich) was added to each well for 15 min to stain biofilm. The excess of unbound dye was removed with MilliQ water. Retained crystal violet was solubilized by the addition of 150 of µl 96% ethanol for 15 min, and absorbance was measured at 570 nm using spectrophotometer (Beckman Coulter). The experiments were performed in six

technical replicates and repeated three times (*Cronobacter malonaticus* 161007/29) or six times (control, Te-R). As an internal control of the biofilm formation methodology, we used the bacterium *Cronobacter malonaticus* 161007/29.

## Auto-aggregation assay

Auto-aggregation assays were performed according to the published method with slight modifications (*Ghane, Babaeekhou & Ketabi, 2020*). Overnight cultures of bacterial cells (Te-R and control) were centrifuged at 6000g for 10 min and the pellets were resuspended 2 ml of in PBS pH $= 7.4$ ($OD_{600} = 0.5$). The cultures were incubated for 24 h at 25 °C and the absorbance was measured at 600 nm using spectrophotometer (Varioskan, ThermoFisher-Scientific) before (0 h) and after (24 h) incubation. The results were represented by two independent experiments performed with three technical replicates. The auto-aggregation percentage was determined as $[(A_0\text{-}A_1)/A_0] \times 100$ where $A_0$ represented the absorbance of the culture at 0 h and $A_1$ represents the absorbance of the culture after 24 h according to the published method (*Wang et al., 2015*).

## Whole-genome sequencing and sequence analysis

Total DNA was isolated by DNeasy Blood & Tissue kit (Qiagen, Hilden, Germany) according to manufacturer's instructions. Totally 1 ng of bacterial DNA quantified fluorometrically with Qubit 2.0 Fluorometer (Thermo Fisher Scientific, Waltham, MA, USA) was used as template for transposon-based library preparation by Nextera XT library preparation kit (Illumina, San Diego, CA, USA), according to standard protocol. After low-cycle indexing pcr amplification with Nextera XT Index Kit (Illumina, San Diego, CA, USA) and purification using Agencourt AMPure XP beads (Beckman Coulter, Brea, USA) for size-selection, libraries were quantified with Qubit dsDNA HS Assay Kits (Thermo Fisher Scientific, Waltham, MA, USA) and the average fragment size was determined using Agilent High Sensitivity DNA Kit (Agilent Technologies, Waldbronn, Germany). Final libraries were pooled together in equimolar ratio and analysed using paired-end ($2 \times 250$) sequencing on Illumina MiSeq platform (Illumina, San Diego, California, USA). Sequencing data were imported into CLC Genomics Workbench Version 9.5.2 (Qiagen). Each sequence of sample was treated by merging and trimming (0.01) and reads shorter than 50 nucleotides were discarded. Trimmed reads were mapped to reference genome of *Escherichia coli BL21(DE3)* (NC_012971). Aligned reads were deduplicated and locally realigned with default settings. Variant calling was performed using fixed ploidy variant detection tool with fixed ploidy 1, required variant probability 90% and minimum frequency 20%.

## Determination of mutation rate

The mutation rate ($\mu$) was determined according *Pope et al. (2008)* by using the equation $\mu = [(r_2/N_2) - (r_1/N_1)] \times \ln(N_2/N_1) = (f_1 - f_2) \times \ln(N_2/N_1)$, where $r_1$ is the observed number of mutants at time point 1, $r_2$ is the observed number of mutants at the next time point, and $N_1$ and $N_2$ are the numbers of cells at time points 1 and 2, respectively, while $f_1$ and $f_2$ are the mutant frequencies at points 1 (which was 0 s) and 2 (which was 120 s). The results were then compared.

## Results and Discussion

We have selected the appropriate strain of *E. coli BL21(DE3)* (NC_012971) according to the availability of the reference genome in the database, to facilitate work with the sequencing data. The bacterial strains and plasmids were obtained from the collection of microorganisms of the Department of Molecular Biology, Comenius University in Bratislava, Faculty of Natural Sciences.

Isolation of the plasmids (pLK18 - *ter+* and pACYC184 - *ter-*) was followed by electroporation into the appropriate strain mentioned above. Consequently, we have prepared Ter-positive cells and Ter-negative cells to help us compare their UVC treatment response. The bacterial resistance and sensitivity to tellurite provided by the above-mentioned plasmids was already proved in our previous work (*Aradska et al., 2013*).

Triplicates of bacterial cultures at Petri dishes were exposed to 0 s (0.0 mJ/cm$^2$), 120 s (300 mJ/cm$^2$) and 1200 s (3,000 mJ/cm$^2$) of UVC dose calculated at the UVC device aperture and plate-lamp distance of 60 cm in laboratory biological safety cabinet. In the stationary phase of growth, neither Ter-positive nor Ter-negative cells survived after 3,000 mJ/cm$^2$ of UVC radiation. Contrarily, Ter-positive cells showed an average of 5-fold higher resistance to UVC radiation compared to Ter-negative cells after 300 mJ/cm$^2$ of UVC exposure. The percentage of survival was based on colony forming unit measurements, of Ter-negative and Ter-positive cells. The percentage of survived Ter-negative cells ($M = 4.00$, SEM $\pm 1,780$, $n = 4$) was smaller than the percentual survival of the Ter-positive group ($M = 20.25$, SEM $\pm 2,287$, $n = 4$). This difference was considered statistically significant, t (6) = 5.60, $p = 0.0007$ (One-tailed) (Fig. 1; Data S1). At an irradiance of 2.5 mW/cm$^2$ calculated according to the lamp aperture, 120 s UVC irradiation time reduced bacterial colony forming units (CFUs) by 79.75% in Ter-positive and by 96% in Ter-negative cells in liquid LB medium. These results indicate that the Ter-negative cells have reduced ability to survive following UVC exposure. We presumed that the minimal *ter* operon (*terBCDE* $\Delta F$) may be involved in increased UVC resistance of *E. coli*, due to the fact that it is responsible for many other resistances such as tellurium, oxidative stress, colicin resistance and is also implicated in phage inhibition. Our assumptions have been partially confirmed in this work, in the context that this was only pilot study.

An analysis of the whole bacterial genome in CLC software enabled to reveal that cells after 120 s (300 mJ/cm$^2$) UVC irradiation show a higher proportion of mutations (insertion and SNV) in Ter-negative cells. In the *cynX* (cyanate MFS transporter) gene, insertion of T occurred at position 330099, in the *glgC* (glucose-1-phosphate adenylyltransferase) gene a single nucleotide variation of G to A occurred at position 343170, but these mutations were missense. No changes were found in Ter-positive cells after irradiation, a deletion present in the cells was present even before irradiation. In the *pcnB* (polynucleotide adenylyltransferase PcnB) gene, deletion of T occurred at position 160821, also this mutation was missense (Table 1). All sequencing data has been deposited in a public database GenBank under the accession no. of BioProject PRJNA655930. This project consists of sequences for Ter-positive cells after 120 s irradiation (BioSample: SAMN15762301; Sample name: 8LT-Bl21-minTer-120s; SRA: SRS7167470), Ter-negative cells after 120 s irradiation (BioSample: SAMN15762280; Sample name: 3LT-Bl21-ctrl-120s; SRA: SRS7167466), Ter-negative cells

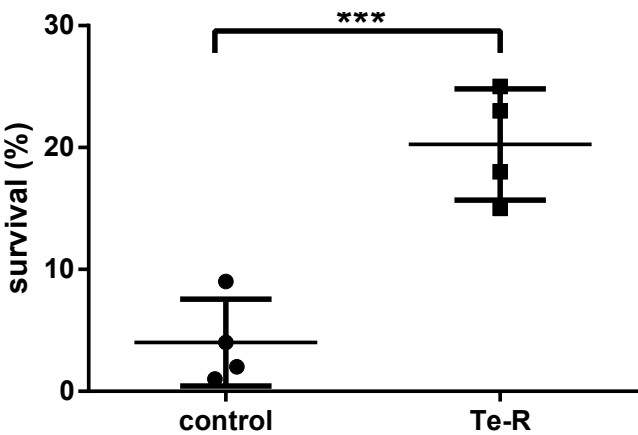

**Figure 1** **Percentual survival after 120 s (300 mJ/cm$^2$) exposure of UVC.** Note that after UV treatment, there was significant difference (t (6) = 5.60, $p = 0.0007$) between the percent survival of the Ter-positive group (Te-R) and Ter-negative group (control). Error values represent the standard deviation of experiments.

after 0 s irradiation (BioSample: SAMN15762266; Sample name: 1LT-Bl21-ctrl-0s; SRA: SRS7167437) and Ter-positive cells after 0 s irradiation (BioSample: SAMN15762300; Sample name: 5LT-Bl21-minTer-0s; SRA: SRS7171211). We determined the mutation rate of the cells after 120 s UVC irradiation using the above equation. The mutation rate of Ter-negative cells was higher (161.72) than the mutation rate of Ter-positive cells (0).

Although this is the first study of the *E. coli* minimal *ter* operon in the role of UVC resistance, studies in *D. radiodurans* are known to reveal the association of Ter proteins with UVC. To understand the functions of individual genes and cellular systems in *D. radiodurans* as well as their relationship with other organisms *Makarova et al. (2001)* undertook a detailed computational analysis of the *D. radiodurans* genome. Expansion of proteins of the TerDEXZ/CABP family in *Deinococcus* is interesting because some of these proteins could confer resistance to a variety of DNA-damaging agents, including methyl methanesulfonate, mitomycin C, heavy-metal cations, and UV (*Azeddoug & Reysset, 1994*; *Jobling & Ritchie, 1988*) and other forms of stress (*Antelmann et al., 1997*). This family si likely to be related to stress response (*Makarova et al., 2001*). In another work, *Karlin & Mrazek (2001)* proposed that Tellurium resistance protein TerD is one of the proteins, that help intrinsically in maintaining the survival and stability of the *D. radiodurans* cell when exposed to severe conditions of UV radiation. Moreover, TerD protein can potentially stand out in the manifold detoxification facilities that neutralize and remove free oxygen radicals and other toxic substances. Their approach was based on predicting gene expression levels related to codon usage differences among gene classes. In this context, *terD* was identified as a highly expressed gene. Further, *Sweet & Moseley (1974)* investigated that in exponential phase, *D. radiodurans* is 33-fold more resistant to UV than is *E. coli*. *Krisko & Radman (2010)* reported that 4% of *E. coli* cells (*E. coli* MG1655 wild type, *E. coli* ΔrecA:: kan, *E. coli* CB1000, *E. coli* CB2000, *E. coli* CB founder strain (MG1655)) survived after irradiation with a dose of approximately 100 J/m$^2$ (1,000 mJ/cm$^2$). It is important to emphasize that

Jánošíková et al. (2021), *PeerJ*, DOI 10.7717/peerj.11197
**Table 1** **Detailed mutation list of the Ter-positive and Ter-negative (control) strain after 120 s (300 mJ/cm$^2$) UVC exposure.**

| Strain | Reference Position | Gene | Type | Length | Reference | Allele | Zygosity | Count | Coverage | Frequency | Forward/reverse balance | Average quality | Overlapping annotations | Coding region change | Amino acid change |
|---|---|---|---|---|---|---|---|---|---|---|---|---|---|---|---|
| **Ter-negative** | 330099 | cynX | Insertion | 1 | – | T | Homozygous | 46 | 89 | 51,68539326 | 0,478261 | 36,76087 | N/A | N/A | N/A |
| **Ter-negative** | 3431701 | glgC | SNV | 1 | G | A | Homozygous | 39 | 75 | 52 | 0,493671 | 35,15385 | N/A | N/A | N/A |
| **Ter-positive** | 160821 | pcnB | Deletion | 1 | T | – | Homozygous | 42 | 42 | 100 | 0,460317 | 35,57143 | N/A | N/A | N/A |

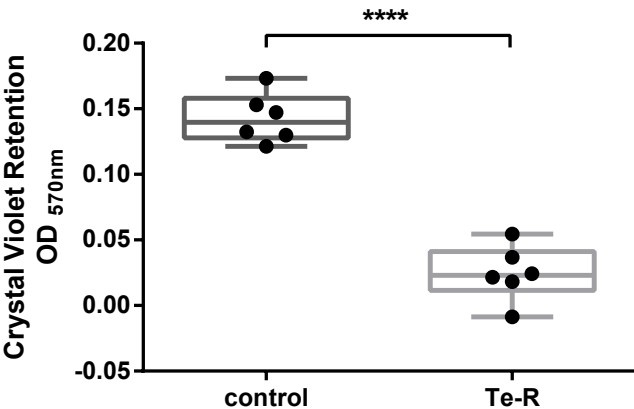

**Figure 2** **Ter-positive group (Te-R) is defective in biofilm formation compared with Ter-negative group (control) depending on Crystal Violet retention.** All statistical analyses were performed in Graph-Pad Prism 6.0 software by unpaired $t$-test. **** $P < 0,0001$.

the strains of bacteria used in the study did not contain *ter* genes. In our study, only 4% of *E. coli* cells survived, that did not contain *ter* genes, but at a much lower dose, the dose of 30 J/m$^2$ (300 mJ/cm$^2$) calculated at the UVC device aperture (plate–lamp distance was 60 cm). This more than three-fold difference between irradiation doses may be due to differences in the design of the experiment. In our case, we irradiated the cells with constant shaking in a liquid medium. Another study identified, that oxidative stress-responsive proteins within tellurium resistance operon TerB (DR2220) and TerD (DR2221) were upregulated in cells of *D. radiodurans* exposed to UVC/vacuum conditions and analysed by an integrative proteometabolomic approach (*Ott et al., 2017*). In context of previous findings, we can assume that *ter* genes do not play a role directly in repairing DNA damage caused by UVC radiation, but rather provide increased resistance to oxidative stress caused by UVC radiation. Thus, they could indirectly provide a reduction of DNA damage.

Furthermore, a significant difference between Ter-positive (Te-R) cells and Ter-negative (control) cells in biofilm formation and auto-aggregation was demonstrated in our study for the first time. We hypothesized that some of given phenotypes could contribute to increased resistance to environmental stressors in Ter-positive (Te-R) cells. We found a deficiency in biofilm formation in Ter-positive (Te-R) cells compared to Ter-negative cells (control) (**** $P < 0.0001$) (Fig. 2) (Data S2). As an internal control of the method of biofilm formation, we used the bacterium *Cronobacter malonaticus* 161007/29 (*C. malonaticus* 161007/29), which is well known to form biofilm on various surfaces (*Ye et al., 2018*). Overall, the mean OD$_{570}$ for *C. malonaticus* 161007/29 was significantly higher than the mean OD$_{570}$ for Ter-positive (Te-R) and Ter-negative cells (control) (**** $P < 0.0001$) (Fig. 3) (Data S3).

In this study, auto-aggregation ability of *E. coli* strains (Ter-positive and control) were investigated. Ter-positive strain showed strong ability of auto-aggregation (96.09%) in comparison to the Ter-negative (control) strain (69.53%) (Data S4). However, the exact mechanism for ensuring auto-aggregation in Ter-positive cells is currently unknown.

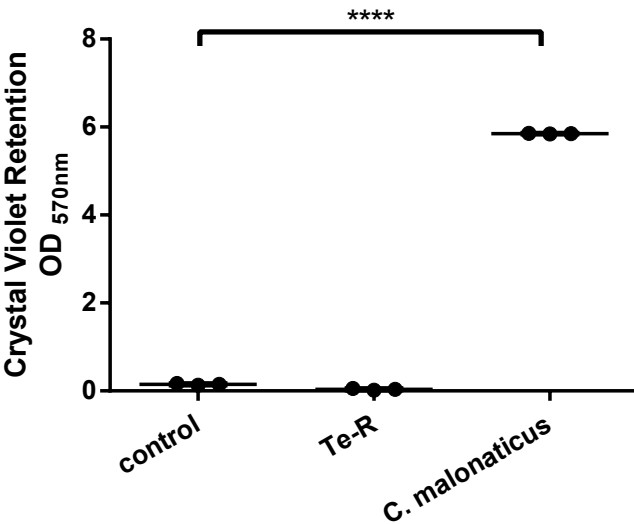

**Figure 3** **Ter-positive group (Te-R) is defective in biofilm formation compared with Ter-negative group (control) and** *Cronobacter malonaticus* **depending on Crystal Violet retention.** All statistical analyses were performed in GraphPad Prism 6.0 software by one-way ANOVA. **** $P < 0.0001$.

The auto-aggregation phenotype of the Ter-positive cells can be a part of defence machinery against environmental stress conditions. The association between auto-aggregation and environmental stress defence was also demonstrated in the experiment of the tellurium resistant *tmp*-expressing *E. coli* cells (*Prigent-Combaret et al., 2012*).

Bacterial biofilm formation is often preceded by the formation of cell auto-aggregates (*Trunk, Khalil & Leo, 2018*). However, there are exceptions where cells form auto-aggregates but do not form a biofilm like in the study of *Hiramatsu et al. (2016)*. In this study the BipA mutant of *Bordetella holmesii* exhibited the strong auto-aggregation phenotype, but failed to form biofilms. *Bordetella* intermediate protein A (BipA) plays a big role in preventing auto-aggregation and indirectly promoting biofilm formation in *Bordetella holmesii*. Another example is *Burkholderia pseudomallei 08*, where auto-aggregation is mediated by *pilA* in a temperature-regulated manner. On the other hand, *pilA* reduces biofilm formation by *Burkholderia pseudomallei 08* (*Boddey et al., 2006*).

It is already well known that various morphological changes (auto-aggregates and/or biofilms) of bacterial cells affect the survival of these microorganisms under stressful conditions (*De Carvalho, 2017*; *Caceres et al., 2014*; *Kostakioti, Hadjifrangiskou & Hultgren, 2013*; *Haaber et al., 2012*; *Anderson & O'Toole, 2008*; *Hall-Stoodley, Costerton & Stoodley, 2004*; *Elasri & Miller, 1999*).

In conclusion, our study shows that the *E. coli* minimal *ter* operon does show an increased ability of cells to survive UVC irradiation and smaller mutation rate after 120 s. We have clearly shown that Ter-positive cells form auto-aggregates to an increased extent but are deficient in biofilm formation under the indicated conditions. From these findings, we can assume that the formation of auto-aggregates provides a physiological advantage to Ter-positive cells in comparison with Ter-negative (control) cells under stress conditions.

However, the exact molecular mechanism of increased UVC resistance and increased auto-aggregation is not known in this case.

## CONCLUSIONS

Our pilot study shows that Ter-positive cells display lower sensitivity to UVC radiation, corresponding to a presence in minimal *ter* operon. Ter-positive cells showed an average of 5-fold higher resistance to UVC radiation compared to Ter-negative cells after 120 s (300 mJ/cm$^2$) of UVC exposure calculated at the UVC device aperture and plate-lamp distance of 60 cm. As expected, we found that neither Ter-positive nor Ter-negative cells survived after 1200 s (3,000 mJ/cm$^2$) of UVC radiation also calculated at the UVC device aperture and plate-lamp distance of 60 cm, which is commonly used to disinfect the laminar box. The role of the *E. coli* minimal *ter* operon in higher UVC resistance is evident. Percentual survival was considered statistically significant, t (6) = 5.60, $p = 0.0007$. At an irradiance of 2.5 mW/ cm$^2$ calculated according to the lamp aperture, 120 s UVC irradiation time reduced bacterial colony forming units (CFUs) by 79.75% in Ter-positive and by 96% in Ter-negative cells in liquid LB medium. The effect of UVC radiation on the DNA damage at the whole genome level exhibited smaller mutation rate in Ter-positive cells than in Ter-negative cells. The Ter-positive strain exhibited 26% higher auto-aggregation activities and were able to inhibit biofilm formation more than Ter- negative strain (**** $P < 0.0001$). Our findings suggest that auto-aggregation ability is related to minimal *ter* operon. We hypothesize that the auto-aggregation phenotype may provide to Ter-positive cells a physiological advantage under the stress conditions.

## ACKNOWLEDGEMENTS

The authors acknowledge Mrs. Andrea Jánošíková's help in proofreading this manuscript. Special thanks for valuable advice as well as help in research to Mgr. Veronika Kadličeková, PhD.

### Funding

This study was supported by the University of Ss. Cyril and Methodius in Trnava grant FPPV-04-2019 and is the result of the project implementation supported by the Research and Development Operational Programme funded by the ERDF (ITMS 26240220086). The funders had no role in study design, data collection and analysis, decision to publish, or preparation of the manuscript.

### Grant Disclosures

The following grant information was disclosed by the authors:
University of Ss. Cyril and Methodius in Trnava: FPPV-04-2019.
ERDF: ITMS 26240220086.

## Competing Interests

Lenka Pálková is employed by Medirex group, Slovak Republic.

## Author Contributions

- Lenka Jánošíková conceived and designed the experiments, performed the experiments, analyzed the data, prepared figures and/or tables, authored or reviewed drafts of the paper, and approved the final draft.
- Lenka Pálková conceived and designed the experiments, performed the experiments, authored or reviewed drafts of the paper, and approved the final draft.
- Dušan Šalát and Andrej Klepanec analyzed the data, authored or reviewed drafts of the paper, and approved the final draft.
- Katarina Soltys conceived and designed the experiments, performed the experiments, analyzed the data, authored or reviewed drafts of the paper, and approved the final draft.

## DNA Deposition

The following information was supplied regarding the deposition of DNA sequences:

All sequencing data are available at GenBank: PRJNA655930.

## Data Availability

Raw data are available in the Supplemental Files.

## Supplemental Information

Supplemental information for this article can be found online at http://dx.doi.org/10.7717/peerj.11197#supplemental-information.

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
