# Peer review of "Response of Escherichia coli minimal ter operon to UVC and auto-aggregation: pilot study"

_PeerJ, doi:10.7717/peerj.11197_

## Round 0.1 · original submission · Major Revisions

Please address all the issues raised by the reviewers carefully which will considerably improve your manuscript.

Reviewer 1 ·

Basic reporting

The subject of the manuscript is interesting and the experiments are well performed. The results are believable. However, the manuscript is, at times, wordy, especially the discussion that must be shortened.

Experimental design

Correct

Validity of the findings

OK

Additional comments

The subject of the manuscript is interesting and the experiments are well performed. The results are believable. However, the manuscript is, at times, wordy, especially the discussion that must be shortened. The Introduction needs more clarifications for the benefit of a general reader. I suggest to substantially modify the manuscript in the sense that it needs a better Introduction and a much shorter Discussion, or even a combination of Results and Discussion. The paragraphs between lines 256 and 284 could be deleted, and also the last phrases from lines 287 (“Investigating…) to the end of the Discussion.
Minor points:
Line 51: Why is a ‘minimal’ operon? Is there any maximal one?
Line 53: Explain what UV irradiation does to the DNA
Line 65: It should be stated from the beginning that the operon is encoded by a plasmid, so that it can be horizontally transferred.
Lines 204-205: Did the survivals resume normal growth after irradiation?
I found some small typos in lines 71, 229, and 245.

Reviewer 2 ·

Basic reporting

The structure of manuscript by Janosikova et al fits in to PeerJ requirements. The introduction demonstrate enough background and literature is relevant. English is plain and acceptable, with some misspellings like on line 243, some inconsistency throughout the text like UVC and UV-C and missed intervals between words like in legend for Fig. 2. Figures are relevant to the content of the article, but not all of them have a good quality. There is not enough explanations for Tables 4 and 5 and it is not clear if they are fully presented because there are no right margins for both of them. Raw data Excel file named “Percentual survival after 120 s (300 mJ/cm2) exposure of UV-C” is provided, but presentation of data in that file is the same as in table 3. Is there any reason to show in a separate file the same numbers as in Table 3? Why Excel was chosen for presenting those eight numbers? There are no calculations or any sorting…

Experimental design

Authors understand (lines 291-292) that standards in the field of radiation resistance is presenting radiation dose-survival curves. Such experiment requires just three days of work and it will validate authors’ conclusions extensively.
There are some major confusions in the description of methods.
1) In the abstract authors report that the large fraction E. coli survives 120-s exposure to UVC light and provide number for radiant exposure 300 mJ/cm2. It sounds like too high radiant exposure for about 20% E. coli survival (please compare with Anita Krisko and Miroslav Radman “Protein damage and death by radiation in Escherichia coli and Deinococcus radiodurans” www.pnas.org/cgi/doi/10.1073/pnas.1009312107 .
Indeed, from Materials and Methods section we can see that radiant exposure was estimated at the UVC device aperture, not at 60 cm distance from UVC lamp where samples were irradiated. So, to avoid the confusion, authors should not use the number 300 mJ/cm2 without explanation about 60 cm distance. Of course, it would be better to measure radiant flux directly where samples were by using UVC light dosimeter.
2) It is unclear how the authors described their calculation of CFU/ml. They irradiated 5 ml of cell culture in a small Petri dish. After UVC radiation they plated 0.01 ml of those cells. It means that CFU/ml = number of colonies at certain dilution x dilution factor x 100. It is not clear why they divide on volume of culture plate (mL).
3) It is not clear where a legend is for Supplemental Figure 1. Some explanations for Suppl. Fig. 1 are needed: why DNA of pLK18 plasmid is almost invisible and why two DNA MW markers are shown?

Validity of the findings

The manuscript adds to the understanding of the role of minimal ter operon to radiation resistance of bacteria. Authors hypothesize that minimal ter operon is important for E. coli UVC resistance. In general, the data presented in the manuscript look novel and interesting. The results are relevant to the hypothesis and support authors’ conclusions. However, their presentation must be improved significantly prior to publication. Importantly, the additional experiments are needed. The authors should provide survival curves for both strains irradiated by UVC. Very important that the authors should use radiant exposure numbers more accurately.

Reviewer 3 ·

Basic reporting

The paper presented results on the UV-survival and DNA mutation rate of E.coli cells complemented with ter operon, genes that proved to give resistance to tellurium and other stress.
Some parts need clarification, but in general, the manuscript is well written.
Nevertheless, the results are little described and poorly exploited. Tables and figures are of bad quality or not appropriated to present the results.
In the discussion, the authors confront previous work and evidence related to the ter operon as a summary or review of prior work. Still, in the end, there is no clear explanation of why the cells that carry the ter operon are more fitted under UV than the ones lacking these systems; maybe complementary assays will be needed to arrive at a more advanced conclusion.

The raw data of sequences are shared in the database.

The literature references are sufficient and indicated the lack of reports and the importance of studying the relationship between UV resistance and ter operon.

Experimental design

The original primary research fits within the Aims and Scope of the journal.
The experimental procedures are relatively simple, UV-exposure, plating, UFC counting, with the addition of detecting mutations by WGS sequencing different exposed colonies with and without the operon.
The methods are described with sufficient detail & information to replicate.
The research question: are ter genes improving/related to UV-resistance in E.coli? is it only answered at a basic level. The authors showed that there is evident more growth after UV when the cells have the ter operon and that this correlates with more mutations in their DNA. But they do not perform other experiments that can help provides clues on why is that: oxidative stress markers, ultrastructural changes, overexpression of repair genes, DNA photoproducts measurement, etc.

Validity of the findings

As I said before, the authors explored the UV resistance due to the occurrence/expression of ter operon in the transformed cells and showed evidence on these regards. But it is not clear the mechanism (lack of further experiments), but there is no clear explanation of this behavior with contrasting data from previous works.
The results are poorly exploited, and the figures/tables are not informative enough.
The manuscript is too repetitive in all its sections.

Additional comments

The authors will need to exploit the results they already have and learn to present them more attractively, especially on the results of the WGS of each genome.
Figure 1 is ok.
Figure 2 is incomprehensible.
Tables 1, 2, and 3 are unnecessary as those results are covered in Fig 1.
Tables 4 and 5 should be combined, improved for clarity, and better explained in the text.

---

## Round 0.2 · Major Revisions

Dear Dr. Janosikova,

Please understand my decision. Although it is very difficult to remain scientifically productive during the COVID-19 pandemic, although our labs are not running as usual, I ask you to perform at least some of the experiments suggested by Reviewer 3. As it now stands, your manuscript is really quite "thin" and preliminary.

Thank you for your understanding

Kind regards,
Elisabeth Grohmann

Reviewer 1 ·

Basic reporting

I found that the authors have addressed properly all the concerns raised by the reviewers, and I find that, in its present form, the manuscript is acceptable.

Experimental design

OK

Validity of the findings

OK

Additional comments

I find that my concerns have been properly addressed.

Reviewer 2 ·

Basic reporting

The manuscript by Janosikova et al fits in to PeerJ requirements.

Experimental design

Authors corrected the mistake in the description of their of CFU/ml.
Accepted.

Validity of the findings

The manuscript by Janosikova adds to the understanding of the role of minimal ter operon to radiation resistance of bacteria. Authors hypothesize that minimal ter operon is important for E. coli UVC resistance. In general, the data presented in the manuscript look novel and interesting. The results are relevant to the hypothesis and support authors’ conclusions.

In my first review I suggested to perform additional experiments building survival curves for both strains irradiated by UVC.
The authors replied that it is not possible in their lab: "If we assumed that cell viability would decrease with increasing radiation (since only 4% of Ter- negative cells and 20% of Ter- positive cells survived after 2 minutes of irradiation), then we would have to reduce the radiation dose. We do not have another UVC emitter available in our laboratory, which would allow us to set the desired doses needed to create a credible survival curve."
The explanation sounds strange to me. I do not see any problem in UVC radiation for periods of time shorter than 2 min. It is easy to turn off UVC light in biological safety cabinet after seconds of UVC exposure.
However, I understand that it is not easy to plan experimental work in the present time of COVID-19 and accept authors refusal.

Additional comments

When situation allows, please consider building full survival curves in your experimental work like presented in this manuscript.

Reviewer 3 ·

Basic reporting

The authors have improved the manuscript, on respect to the structure and general discussion. But now, the manuscript results are only one table and one growth figure.
As the authors say in the text the work is a pilot study, for me this is too preliminary and incomplete to be published at this stage.

Experimental design

The authors did not perform any new experiment suggested by the reviewers

Validity of the findings

The work is incomplete as it is presented.

---

## Round 0.3 · Minor Revisions

The reviewers and I highly acknowledge the improvements you made to the manuscript by adding two new experiments and modifying the text accordingly. Your manuscript has significantly improved by these amendments.
However, there are still some points to consider:
- Biofilm assay: Include information on the strain of C. malonaticus you applied as a control (strain name and origin)
"The experiments were performed in six technical replicates and repeated three or six times." Please be more precise: Were they repeated three or six times AND if the number of repetitions differs, then please detail which assay was repeated three times and which six times
- Auto aggregation assay: Please include information on how many technical replicates you performed.
- The scientific English in the new sections should be revised by a fluent speaker to facilitate reading and understanding of the text.

---

## Round 0.4 · accepted · Accept

Congratulations! Now it is a nice and interesting manuscript.

Best regards,
Elisabeth Grohmann